# Enhanced Oral Absorption of Icaritin by Using Mixed Polymeric Micelles Prepared with a Creative Acid-Base Shift Method

**DOI:** 10.3390/molecules26113450

**Published:** 2021-06-06

**Authors:** Cheng Tang, Xiaoming Chen, Hua Yao, Haiyan Yin, Xiaoping Ma, Mingji Jin, Xin Lu, Quntao Wang, Kun Meng, Qipeng Yuan

**Affiliations:** 1College of Life Science and Technology, Beijing University of Chemical Technology, Beijing 100029, China; tczy033@126.com; 2Beijing Shenogen Pharmaceutical Co., Ltd., Beijing 102206, China; xiaoming.chen@shenogen.com (X.C.); hua.yao@shenogen.com (H.Y.); Haiyan.yin@shenogen.com (H.Y.); xiaoping.ma@shenogen.com (X.M.); mingji.jin@shenogen.com (M.J.); xin.lu@shenogen.com (X.L.); quntao.wang@shenogen.com (Q.W.); kun.meng@shenogen.com (K.M.)

**Keywords:** acid-base shift method, icaritin, polymeric micelles, oral absorption, transmembrane transport

## Abstract

The purpose of this study was to develop mixed polymeric micelles with high drug loading capacity to improve the oral bioavailability of icaritin with Soluplus^®^ and Poloxamer 407 using a creative acid-base shift (ABS) method, which exhibits the advantages of exclusion of organic solvents, high drug loading and ease of scaling-up. The feasibility of the ABS method was successfully demonstrated by studies of icaritin-loaded polymeric micelles (IPMs). The prepared IPMs were characterized to have a spherical shape with a size of 72.74 ± 0.51 nm, and 13.18% drug loading content. In vitro release tests confirmed the faster release of icaritin from IPMs compared to an oil suspension. Furthermore, bioavailability of icaritin in IPMs in beagle dogs displayed a 14.9-fold increase when compared with the oil suspension. Transcellular transport studies of IPMs across Caco-2 cell monolayers confirmed that the IPMs were endocytosed in their intact forms through macropinocytosis, clathrin-, and caveolae-mediated pathways. In conclusion, the results suggested that the mixed micelles of Soluplus^®^ and Poloxamer 407 could be a feasible drug delivery system to enhance oral bioavailability of icaritin, and the ABS method might be a promising technology for the preparation of polymeric micelles to encapsulate poorly water-soluble weakly acidic and alkaline drugs.

## 1. Introduction

Icaritin (Scheme 1A), a prenylflavonoid derivative from the *Epimedium* genus that has been historically used in Traditional Chinese Medicine, is considered to exhibit various potential pharmacological and biological activities such as anti-cancer (in Phase III clinical trial [1]), anti-inflammation [2], anti-osteoporosis [3], and treatment of erectile dysfunction [4]. Due to its rigid structure, the aqueous solubility of icaritin is less than 1 μg/mL. In addition, the membrane permeability coefficient of icaritin is only about 10^−8^ cm/s probably resulting from its extreme hydrophobicity, resulting in a very unfavorable oral bioavailability of 2% which severely limits its clinical application. To improve the absorption of icaritin, a number of approaches have been applied to enhance the solubility and permeability. For example, Li et al. [5] prepared nanocrystals of icaritin with a size of about 220 nm through the antisolvent-precipitation method. However, the oral bioavailability was only enhanced two-fold compared with icaritin suspension. Our previous study in dogs showed that the oral bioavailability of an oil suspension of icaritin was about 2%. Therefore, there has been an urgent request to develop a vehicle that can efficiently deliver icaritin across the intestinal epithelium.

Recently, nano-drug delivery systems have presented great potential to overcome the intestinal epithelium barrier for poorly absorbed drugs. Among these, polymeric micelles have attracted much attention [6,7,8]. Polymeric micelles with a core-shell structure and sizes ranging from 10 to 100 nm are self-assembled from amphiphilic polymers. It has been reported that polymeric micelles can increase the solubility and permeability and, in turn, oral bioavailability of poorly water-soluble drugs [9,10]. The free amphiphilic polymer itself existing in polymeric micelles solution was documented to have the ability to inhibit P-glycoprotein efflux [11]. Consequently, polymeric micelles can promote cellular uptake and transcytosis of drugs loaded in them across the intestinal epithelium [8,12]. Therefore, polymeric micelles are a promising oral delivery system for poorly water-soluble and permeable drugs.

Currently there are two major classes of methods to prepare drug-loaded polymeric micelles according to the hydrophilic-hydrophobic property of the micelle-forming polymers [13]. As depicted in Scheme 2, the first class of methods, which is usually applicable to intermediate hydrophobic polymers such as poloxamers [14], includes the simple equilibrium method (Scheme 2C) [15] and the oil-in-water emulsion method (Scheme 2D) [16]. The main advantages of the simple equilibrium method are simplicity, rapidity, exclusion of organic solvents (often) and ease of scaling-up, all of which are beneficial to the development of a pharmaceutical formulation. Unfortunately, this method is not suitable for poorly water-soluble drugs. As for the oil-in-water emulsion method, the use of organic solvents is a fatal weakness. The second class of methods for micelle preparation, applied for not readily water-soluble amphiphilic polymers, includes the thin film hydration method (Scheme 2A) [17] and the dialysis method (Scheme 2B) [6]. Although the superiority of these methods to the first class is the high drug loading content achieved, their disadvantages are still apparent: the use of organic solvents, complexity and poor feasibility of scaling-up. Nevertheless, the drug loading content makes it difficult to meet the dosing requirement. Therefore, an efficient and feasible method for preparation of drug-loaded polymeric micelles should be developed for industrialized production.

In this study, we developed an acid-base shift (ABS) method (Scheme 2E) for preparing icaritin-loaded polymeric micelles (IPMs) based on Soluplus^®^/Poloxamer 407, and its feasibility was validated with different properties of water-soluble drugs (Scheme 1B–D). The prepared IPMs were characterized in their physicochemical properties with regard to the morphology, size and its distribution, drug loading, stability under storage and in gastrointestinal fluid, and in vitro release behavior. The in vivo pharmacokinetics in beagle dogs were evaluated in comparison with oil suspension of icaritin. Additionally, the transcellular transport mechanisms for IPMs were disclosed.

## 2. Materials and Methods

### 2.1. Materials

Icaritin (purity ≥ 98.5%) and remdesivir (purity ≥ 98.5%) were provided by Beijing Shenogen Pharmaceutical Co. Ltd. (Beijing, China). Diosmin (97% purity), ibuprofen (purity ≥ 99%), dimethylsulfoxide-*d^6^* (*d^6^*-DMSO)*,* D_2_O, Cy5 NHS and Cy5.5 NHS were purchased from Beijing Inno-chem Co. Ltd. (Beijing, China). Poloxamer 407 (P407) and Soluplus^®^ were obtained from Beijing Fengli Jingqiu Pharmaceutical Co. Ltd. (Beijing, China). Chlorpromazine hydrochloride (CMZ), amiloride, genistein, methyl-*β*-cyclodextrin (M*β*CD) and *β*-glucurondase were gained from Sigma-Aldrich (St. Louis, MO, USA). Hank’s balanced salt solution (HBSS), Dulbecco’s modified Eagle’s medium (DMEM), penicillin–streptomycin, trypsin-EDTA were all purchased from Beijing M&C Gene Technology (Beijing, China). Fetal bovine serum (FBS) was supplied by GIBCO (Burlington, Ontario, Canada). Plastic culture flasks (25 and 75 cm^2^) and Transwell 12-well plates were purchased from Corning (Corning, NY, USA). All other chemicals used were of the highest purity available.

### 2.2. Determination of Critical Micelle Concentration

The critical micelle concentration (CMC) for the mixture of Soluplus^®^/P407 was evaluated by ultraviolet (UV) spectrophotometry as reported earlier [18]. Briefly, potassium iodide (KI, 1.0 g) and iodine (I_2_, 0.5 g) were dissolved in deionized water (50 mL) to obtain a KI/I_2_ solution. The KI/I_2_ solution (5 μL) was added into the aqueous solution (15 mL) of Soluplus^®^/P407 (mass ratio: 2:1) with polymer concentrations ranging from 0.00024 to 0.125 mg/mL. Then, the mixtures were vortexed and incubated in the dark at room temperature for 12 h. The absorbance was determined at 366 nm by UV spectrophotometer (UV759CRT, Shanghai Yoke, Shanghai, China). The CMC was determined from the onset of a sharp rise in absorbance plotted versus the logarithm of the polymer concentration.

### 2.3. Preparation of Drug-Loaded Polymeric Micelles

#### 2.3.1. Preparation of Weakly Acidic Drug-Loaded Polymeric Micelles

The weakly acidic drugs icaritin (Scheme 1A), diosmin (Scheme 1B) and ibuprofen (Scheme 1C) were encapsulated in polymeric micelles by the ABS method. In brief, the weakly acidic drug (2.40 g for icaritin, 1.20 g for diosmin and ibuprofen) was dissolved in 0.2 mol/L sodium hydroxide aqueous solution (120 mL) to obtain the base solution of the drug. Soluplus^®^ (9.6 g) and P407 (4.8 g) were dissolved in 0.05 mol/L hydrochloric acid aqueous solution (480 mL) to acquire the acid solution with the polymers. Then the base solution was added into the acid solution dropwise under magnetic stirring (1000 rpm) followed by centrifugation for 20 min at 10,000 rpm to gain the weakly acidic drug-loaded polymeric micelles solution. The ultrafiltration (polyethersulfone membrane, 100 KD) could be used to concentrate the resultant polymeric micelles solution.

#### 2.3.2. Preparation of Weakly Alkaline Drug-Loaded Polymeric Micelles

Remdesivir (Scheme 1D) was selected to be the representative weakly alkaline drug. Remdesivir-loaded polymeric micelles were prepared via the ABS method. In short, remdesivir (1.11 g), Soluplus^®^ (4.0 g), P407 (2.0 g) and concentrated hydrochloric acid (1.7 mL) were all dissolved in purified water (100 mL). A certain volume of the sodium hydroxide aqueous solution (8.0 mol/L) was then added into the resultant acid solution dropwise under magnetic stirring (1000 rpm) to adjust the pH of the solution to be 5.0. The Remdesivir-loaded polymeric micelles solution was obtained following centrifugation for 20 min at 10,000 rpm.

#### 2.3.3. Pilot Production of IPMs

Icaritin (160 g) was dissolved in 0.2 mol/L sodium hydroxide aqueous solution (8 L) to obtain the base solution of the drug. Soluplus^®^ (640 g), P407 (320 g) and 0.05 mol/L hydrochloric acid aqueous solution (32 L) were added to a glass reactor (50 L) and stirred (1000 rpm) for 2 h. Then, the base solution was pumped into the acid solution over 30 min under stirring (1000 rpm) at 20 °C. IPMs were obtained by centrifugation for 20 min at 10,000 rpm.

#### 2.3.4. Preparation of Forster Resonance Energy Transfer (FRET) Micelles

Fluorophore-labelled Icaritin−Cy5 and Icaritin−Cy5.5 were synthesized (Appendix A), and FRET micelles were prepared by ethanol the injection-dialysis method. Briefly, Icaritin−Cy5 (0.5 mg), Icaritin−Cy5.5 (0.5 mg) and icaritin (9.0 mg) were dissolved in ethanol (1 mL) to obtained the drug solution, and Soluplus^®^ (40 mg) and P407 (20 mg) were dissolved in deionized water (8 mL) to gain the aqueous phase. The drug solution was then injected into the above aqueous phase under stirring at 1000 rpm. The FRET micelles were finally obtained by dialysis (MWCO: 3500 Da) against deionized water for 48 h to remove residual ethanol.

### 2.4. Characterization of Drug-Loaded Polymeric Micelles

The mean particle size and its distribution, and the Zeta potential of drug-loaded polymeric micelles were measured by dynamic light scattering (DLS, Nano ZS, Malvern Instruments, Malvern, UK) at 25 °C. All the samples were diluted with deionized water before measurement and then measured in triplicate.

The morphology of the drug-loaded polymeric micelles was observed using a TESCAN MAIA3 field emission scanning electron microscope (SEM, TESCAN, Brno, Czech Republic). A drop of FRET micelles without dilution was placed on a carbon-coated copper grid and air dried. The samples were stained with 2% phosphotungstic acid and observed using a transmission electron microscope (TEM, JEM-1230, JEOL, Tokyo, Japan).

X-ray powder diffraction (XRPD) was used to investigate the crystallography characteristics of the lyophilized drug-loaded polymeric micelles compared with the raw drug, the mixture of the raw drug and polymers, and blank polymeric micelles. An X-ray diffractometer (Ultima III, Rigaku, Japan) with Cu−Ka radiation was utilized, and all scans were administrated with steps of 0.02° at a rate of 4°/min over a 2θ range of 3–40°.

The interaction between the loaded drug and the polymers was investigated by ^1^H-NMR (400 MHz, Bruker AMX−400, Germany).

### 2.5. Determination of Encapsulation Efficiency and Drug Loading Content

To determine the encapsulation efficiency (EE) and drug loading content (LC) of the micelles, the lyophilized micelles powder was accurately weighed and dissolved in methanol. The resulting solution was filtered through a syringe filter (0.45 μm), and the content of the loaded drug in the micelles was determined using a high-performance liquid chromatography (HPLC) system (Agilent 1260 series, Santa Clara, CA, USA) with a UV detector set at a wavelength of 273 nm for icaritin, 254 nm for remdesivir, 275 nm for Diosmin, and 254 nm for ibuprofen. The chromatographic separation of drugs in 20 µL of injected samples was performed on a SunFire C18 column (4.6 × 250 mm, 5 μm, Waters, Worcester County, MA, USA) at 35 °C. Eluting of the mobile phase, for example for icaritin, consisted of solvent A (0.1% phosphoric acid solution) and solvent B (the mixture of methanol and tetrahydrofuran at a ratio of 35:26 (*v*/*v*), containing 0.1% phosphoric acid) in the following gradient program: the mobile phase was initially composed of 50% solvent B, the content of solvent B was linearly increased to 62% from 10 min to 45 min, then reduced to 50% solvent B from 45.1 min to 55 min. All the samples were measured in duplicate. The EE and LC were calculated using the following Equations (1) and (2):(1)$$\mathrm{EE}\;(\%)=\frac{\mathrm{Mass}\;\mathrm{of}\;\mathrm{icaritin}\;\mathrm{loaded}\;\mathrm{in}\;\mathrm{micelles}}{\mathrm{Mass}\;\mathrm{of}\;\mathrm{icaritin}\;\mathrm{fed}\;\mathrm{initially}}\;\times \;100,$$
(2)$$\mathrm{LC}\;(\%)=\frac{\mathrm{Mass}\;\mathrm{of}\;\mathrm{icaritin}\;\mathrm{loaded}\;\mathrm{in}\;\mathrm{micelles}}{\mathrm{Total}\;\mathrm{mass}\;\mathrm{of}\;\mathrm{micelles}}\;\times \;100.$$

### 2.6. Assessment of the Stability of Micelles

To test the storage stability of the icaritin-loaded micelles (IPMs), the freshly prepared IPMs were transferred into glass vials, and stored at 4 °C and 25 °C for 1, 2, 4, and 8 weeks. The particle size and polydispersity index (PDI) and LC were measured at predetermined time points, as described earlier.

The stability of the IPMs in simulated gastric fluid (SGF) and simulated intestinal fluid (SIF) was also assessed based on the changes in the particle size and the PDI and LC of the micelles over time during their incubation in SGF and SIF, under shaking at a speed of 100 rpm at 37 °C. The particle size and PDI, and LC were determined at 1, 2, 4, 6, and 12 h, as described earlier.

### 2.7. Evaluation of In Vitro Release of the Micelles

The in vitro release behavior of icaritin from the IPMs was evaluated using a dialysis-bag diffusion method, as reported previously, with little modification [9]. Briefly, the solution of IPMs (1.0 mL, 1.0 mg/mL) was introduced into a dialysis bag with molecular weight cutoff of 3500 D (Jingke Inc., Beijing, China). The sealed dialysis bag was immersed into 50 mL of SIF or SGF containing 40% ethanol, to guarantee pseudo-sink conditions, and was kept in a shaker at 37 °C and 100 rpm. At fixed time intervals, 1.0 mL of the release medium was taken and replaced immediately with an equal volume of fresh medium. The content of icaritin in the medium was determined by the HPLC method, as described earlier. The oily suspension of icaritin (0.1 mL, containing 1.0 mg of icaritin) mixed with SIF (0.9 mL) or SGF (0.9 mL) was used as control.

### 2.8. Bioavailability Study

#### 2.8.1. In Vivo Experiments

All animal experiments were performed with the approval of Institutional Animal Care and Use Committee of Pharmaron Lab Animal Research in an AAALACi-accredited facility (Certification No.: 001760, Date: 4 November 2019). Male beagle dogs (approximate 10 kg each) were kept under standard conditions (temperature: 23 ± 2 °C, moisture: 55 ± 10%, and a controlled 12 h light/dark cycle) with free access to food and water. Dogs were fasted overnight prior to the experiments. Thirteen dogs were randomly assigned to three treatment groups. Three dogs were administrated intravenously (IV) icaritin solution (2 mg/mL) which consisted of 10% (*v*/*v*) DMSO, 20% (*v*/*v*) Solutol HS15 and 70% (*v*/*v*) saline solution at a dose of 2 mg/kg. Four dogs were given IPMs intragastrically (aqueous solution, 20 mg/mL) at a dose of 20 mg/kg. Six dogs were administered orally the oily suspension of icaritin (soft capsule, 100 mg/capsule) used in the clinic at a dose of 20 mg/kg. Blood samples (1 mL) from the IV group were collected from venipuncture of peripheral veins at 0, 0.083, 0.167, 0.5, 1, 2, 4, 6, 8, 12, and 24 h following the single administration in heparinized polypropylene tubes. Blood samples (1 mL) from the oral group were collected from venipuncture of peripheral veins at 0, 0.167, 0.5, 1, 2, 4, 6, 8, 12, and 24 h post-dose in heparinized polypropylene tubes. All the collected blood samples were immediately centrifuged at 2000× *g* for 10 min at 28 °C, and the plasma samples were then stored at −80 °C until analysis.

#### 2.8.2. Blood Sample Analysis

The plasma (55 μL) was incubated with acetic acid (0.1 mol/L, 10 μL) and phosphate buffered saline (pH 5.0, 25 μL) containing *β*-Glucuronidase (25 μL, 10,000 U/mL) for 1 h at 37 °C. The internal standard solution (200 μL, containing 500 ng/mL of dexamethasone) was then added and vortexed for 2 min. The resultant mixture was centrifuged at 4000× *g* for 15 min, and the 5 μL aliquot of supernatant was subjected to LC/MS/MS analysis using an UHPLC-MS/MS system consisting of an ultra HPLC system (LC-30D, Shimadzu, Japan) coupled with a LC/MS/MS mass spectrometer (AB API 5500, Agilent, Palo Alto, CA, USA) equipped with electrospray-ionization (ESI) source. The chromatographic separation was carried out on a Gemimi C18 (3 μm, 100 A, 50 × 2.1 mm, Phenovmenex, Torrance, CA, USA) column at the column temperature of 40 °C using a mobile phase consisting of 5% acetonitrile in 0.1% formic acid (A) and 95% acetonitrile in 0.1% formic acid (B) pumped at a flow rate of 0.6 mL/min. The program of gradient eluent was set as follows: 0–0.2 min with 20% B, 0.2–1.9 min with 20–100% B, 1.9–2.1 min with 100% B, and 2.1–2.5 min with 20% B. The eluent was directly introduced into an ESI interface. The ESI parameters were set as follows: the temperature of the drying gas (nitrogen), 400 °C; Curtain gas, 45 psi; nebulizer pressure, 50 psi; ion spray voltage, 4500 V for negative ion mode. Multiple-reaction monitoring detection was employed using nitrogen as the collision energy. The icaritin and dexamethasone (internal standard) were monitored at *m*/*z* 367.12–297.00 and *m*/*z* 391.14–361.20, respectively. The linearity of the calibration curve was excellent (R = 0.9957) within the concentration range from 1 to 1000 ng/mL. The coefficient of variation of the intra- and inter-day precisions (relative standard deviation, RSD) of the quality control samples were lower than 15% and the accuracy was in the range of (87.10 ± 0.46)–(107.0 ± 0.05)%. The limit of quantification (LOQ) was 1 ng/mL.

For the IV dosing group, pharmacokinetic parameters were calculated using two compartment analysis with the WinNonlin (Phoenix, version 6.1) program. The main pharmacokinetic parameters were calculated. The area under the blood drug concentration-time curve from 0 to *t* (AUC_0–t_) was calculated with the trapezium method. Total clearance (CL), and mean residence time from 0 to *t* (MRT_0–t_) were obtained.

For the oral dosing groups, the main pharmacokinetic parameters were obtained from blood drug concentration–time data. The AUC_0–t_ was estimated by using the noncompartmental model in the WinNonlin (Phoenix, version 6.1) program.

### 2.9. Cell Culture

The human colon adenocarcinoma cells, Caco-2 cells, at passage 25, were supplied by the Cell Culture Center of Institute of Basic Medical Sciences, Chinese Academy of Medical Sciences and Peking Union Medical College (Beijing, China). The cells were cultured in DMEM medium with 100 U/mL penicillin, 100 mg/mL streptomycin, 1% nonessential amino acids, and 10% FBS at 37 °C in a humidified atmosphere containing 5% CO_2_, and sub-cultivated every 2 days at 80–90% confluence and digested with trypsin-EDTA at a split ratio of 1:3.

### 2.10. Assessment of the Micelle Integrality after Their Transmembrane Transport

The integrity of the micelle structure after their transcytosis transport was identified using the FRET method [8]. In brief, the donor Cy5 and the acceptor Cy5.5 were first conjugated with icaritin to obtain fluorophore-labelled icaritin-Cy5 and icaritin-Cy5.5 (Appendix A Appendix A), respectively. Icaritin, icaritin-Cy5 and icaritin-Cy5.5 were encapsulated in the micelles as described earlier to yield FRET micelles. To examine the existence of FRET, the fluorescence spectrum of FRET micelles in water and in methanol was recorded at an excitation wavelength of 635 nm, with microplate reader (SpectraMax i3, Molecular Devices, San Jose, CA, USA). The emission was scanned from 660 to 750 nm.

Caco-2 cell monolayers were obtained by incubation of 500 μL of Caco-2 cell suspension (2 × 10^5^ cells/mL) on Transwell^®^ polycarbonate membrane inserts with a pore diameter of 0.4 μm (12-well plates, article number: 3460, Corning, Kennebunk, ME, USA) in a CO_2_ incubator for about 21 days [8]. The transepithelial electrical resistance (TEER) was monitored with an electrical resistance meter (Millicell ERS-2, Millipore, Darmstadt, Germany) to detect the integrity of cell monolayers. The monolayers with TEER values of higher than 300 Ω/cm^2^ were used in the transcytosis tests. Subsequently, following removement of the culture medium in the apical (AP) and basolateral (BL) sides, prewarmed HBSS were appended in both sides and incubated at 37 °C for 30 min. 0.5 mL of prewarmed FRET micelles solution with 2.25 μg/mL of icaritin-Cy5 and icaritin-Cy5.5, respectively, in HBSS was then added into the AP side, and 1.5 mL of prewarmed fresh HBSS was added into the BL side. After 4 h of transcytosis, all the medium was collected from the BL side, subjected to the fluorescence spectra recordation in water and in methanol, and TEM examination, as described earlier. In addition, the integrity of the cell monolayer was monitored by TEER during the entire experiment.

### 2.11. Exploration of Transport Mechanisms of IPMs across Caco-2 Cell Monolayers

The transcytosis pathway of IPMs across Caco-2 cell monolayers was studied as reported previously with a little modification [8]. In short, the Caco-2 cell monolayers were pre-incubated with 0.5 mL of HBSS containing four kinds of endocytosis inhibitors such as genistein (100 μM), chlorpromazine hydrochloride (CMZ, 10 μg/mL), amiloride (100 μM), and methyl-*β*-cyclodextrin (M*β*CD, 10 mM) at 37 °C for 30 min. Afterwards, the HBSS in the AP side was replaced with prewarmed IPMs solution (1.0 mg/mL) in HBSS containing the corresponding inhibitors, and then incubated for another 4 h at 37 °C. The medium collected from BL side was treated with 1% acetic acid solution (10 μL) and *β*-glucurondase (45 μL) followed by incubation for 1 h at 37 °C, lyophilized and redissolved in methanol (175 μL). The amount of transported icaritin was determined by HPLC as described earlier.

### 2.12. Statistical Analysis

Two-tailed Student’s *t*-test was employed to perform statistical analysis. A *p* value of ≤0.05 was considered to be statistically significant.

## 3. Results and Discussion

### 3.1. Critical Micelle Concentration Determination

In our preliminary tests, a number of amphiphilic polymers were used to prepare icaritin-loaded polymeric micelles, and we found that Soluplus^®^/P407 mixed micelles had the advantages of high drug loading capacity and favorable stability. P407 with a molecule weight of 9840–14,600 is a triblock copolymer which consists of a central hydrophobic block of polypropylene glycol (PPG) flanked by two hydrophilic blocks of polyethylene glycol (PEG). In contrast, Soluplus^®^ with a molecule weight of 90,000–140,000 is a graft copolymer with a hydrophilic chain of PEG grafted with hydrophobic chain of polyvinyl caprolactam (PVCap) and polyvinyl acetate (PVAc). Possibily due to the huge difference in the length of hydrophobic chains, P407 could stabilize the icaritin-loaded Soluplus^®^ micelles, especially at the mass ratio of 2:1 for Soluplus^®^ and P407. Therefore, Soluplus^®^ and P407 were chosen to prepare the micelles for solubilizing poorly water-soluble molecules.

A lower CMC value of the micelle-forming amphiphilic polymers endows the micelles with better dilution stability in the gastrointestinal tract and in blood circulation. The CMC value lower than 135 mg/L for the micelles is generally considered to impart the ability to resist rapid dissociation after oral administration [19]. In this study, UV-Vis spectrophotometry was used to measure the CMC of Soluplus^®^/P407 mixture in a mass ratio of 2:1 based on the spectral characteristic of donor–acceptor complexes between I_2_ and nonionic surfactants in an aqueous medium [20]. As presented in Figure 1, the CMC was determined to be 7.9 mg/L, which was a little higher than that of Soluplus^®^ (7.0 mg/L, [21]) and much lower than that of P407 (ca. 28 mg/L, [22]). As described below, the concentration of Soluplus^®^/P407 in concentrated IPMs was approximately 120 g/L, implying that the concentrated IPMs could withstand a dilution factor of at least 10,000-fold in an aqueous environment, and thereby keeping icaritin encapsulated inside the micelles in the systemic circulation [23,24].

### 3.2. Preparation of Drug-Loaded Micelles by the ABS Method

In the present research, a creative and ingenious approach (Scheme 2E) to prepare polymeric micelles for loading weakly acidic and alkaline drugs was developed with higher drug loading and exclusion of organic solvents. More importantly, this technology was feasible for scaling-up. A schematic illustration of this method and the experimental process were depicted in Scheme 3.

Firstly, icaritin was encapsulated in the micelles. Icaritin, a flavonoid aglycone, has extremely poor solubility in neutral and acidic solutions; however, it can be dissolved in a base solution in salt form owing to its phenolic hydroxyl group (Ar-OH). Based on the weak acidity of icaritin, icaritin was dissolved in NaOH aqueous solution, and then the solution of Soluplus^®^/P407 in the mass ratio of 2:1 in HCl solution was added and the pH of the mixture was adjusted to be neutral (pH 6.5–7.0) with HCl. During this procedure, the soluble salt of icaritin was converted to free icaritin and was encapsulated in the hydrophobic core of the micelles at the same time. The EE and LC of the micelles were listed in Table 1, which was remarkably higher than those of the micelles prepared by the thin film hydration method (about 5.45% and 0.91%). This novel method for preparing micelles was named the ABS method (Scheme 2E). A schematic illustration of the self-assembly for icaritin-loaded micelles was presented in Scheme 4.

In order to determine whether the icaritin was encapsulated inside the micelles, ^1^H-NMR spectroscopy was performed for the IPMs [25]. Icaritin, blank micelles, and the physical mixture of icaritin and blank micelles were also analyzed as control. As shown in Figure 2, all the proton peaks of icaritin in DMSO (Figure 2A) and the polymers in DMSO (Figure 2D) were clearly observed. However, as with the blank micelles in D_2_O (Figure 2E), only the methylene proton peak (3.59 ppm) of PEG was found for IPMs in D_2_O (Figure 2C), and all the proton peaks of icaritin and the hydrophobic chains of the polymers had disappeared. These findings reinforced the conclusion that icaritin was encapsulated inside the hydrophobic core formed by PPG and PVCap-PVAc and surrounded by hydrophilic shell formed by PEG.

In addition, the stability of icaritin during the loading process was also examined. The HPLC chromatograms of raw icaritin and the loaded icaritin in the micelles are shown in Figure 2F. The purity of raw icaritin and the IPMs was calculated to be 99.7% and 99.5%, respectively. Moreover, each single IPMs’ impurity content of was no more than 0.2%. Thus, the drug degradation induced by ABS process was considered to be negligible according to International Council for Harmonisation of Technical Requirements for Pharmaceuticals for Human Use (ICH) Q3B.

Secondly, in order to validate the feasibility of applying the ABS method to other drugs, we tried to encapsulate three drugs with different functional groups widely existing in drug molecules (Scheme 1) into polymeric micelles with ABS method. Diosmin is used to treat hemorrhoids [26], ibuprofen is a traditional anti-inflammatory drug, and remdesivir effectively inhibits Ebola virus and 2019-nCov replication [27,28]. The first two drugs with phenolic hydroxyl (Ph–OH, Scheme 1B) or carboxyl groups (–COOH, Scheme 1C) are weakly acidic and the last one with amino groups (Scheme 1D) is weakly alkaline. As shown in Table 1, these three drug-loaded polymeric micelles exhibited nano-scaled size (about 63–99 nm) with a narrow distribution (PDI < 0.2), and the encapsulation efficiency (EE) was at least 88%. Quite expectedly, a high drug loading content (LC) was achieved.

In addition, to investigate the scale-up effect of ABS method, large scale IPMs (40 L) were produced in a reactor. Mean size, PDI, EE%, and LC% of the large scale IPMs were similar to those of small scale IPMs. These results revealed that ease of scaling-up was an advantage of the ABS method.

Based on these results, it could be concluded that the ABS method is an alternative technology (Scheme 2E) to prepare micelles for some weakly acidic and alkaline drugs. Unfavorably, this method also had its own insuperable limitation that the micelle-forming amphiphilic polymers were required to be water-soluble.

### 3.3. Characterization of IPMs

The appearance of the solution of IPMs prepared by the ABS method was observed. As presented in Figure 3A, the solution with 3.6 mg/mL of icaritin for the fresh prepared IPMs (left) was a typical colloidal solution with a pale yellow and opalescent appearance and the Ding Daer phenomenon was evident. The concentrated IPMs solution (right) with dark yellow and 20 mg/mL of icaritin obtained via ultrafiltration was similar.

The morphology of IPMs was intuitively observed under Scanning electron microscope (SEM). As shown in Figure 3B, IPMs were well-defined spheroids with relatively uniform diameter. The diameter detected by DLS was 72.74 ± 0.51 nm with a narrow distribution (PDI: 0.176 ± 0.010) (Figure 3C). In addition, the Zeta potential of IPMs was −0.02 mV (Figure 3D), implying that the IPMs were basically electrically neutral.

To examine the physical state of icaritin in IPMs, XRPD analysis was performed. Figure 3E showed the X-ray diffractogram of icaritin, blank micelles, the physical mixture of icaritin and polymers, and IPMs. Raw icaritin showed a high intensity and sharp crystalline peak at 2θ of 6°, indicative of the crystalline nature of icaritin. The characteristic diffraction peak for raw icaritin, which was clearly observed in the diffractogram of the physical mixture of icaritin and polymers, completely disappeared in the diffraction patterns of IPMs which were similar to those of blank micelles. These results further proved that icaritin was encapsulated in the inner core of the micelles. On the other hand, XRPD results strongly suggested that the crystallinity of icaritin loaded inside the nano-sized polymeric micelles was completely demolished, meaning that encapsulated icaritin existed in an amorphous form inside the micelles, which is in agreement with previously published results [29]. This noncrystalline nature of the loaded drug in the polymeric micelles is beneficial to increase drug solubility, facilitate drug release and dissolution, and may enhance the oral absorption of the loaded drug, which is desirable [30].

### 3.4. Stability of IPMs

Physicochemical stability is a crucial for any drug formulation. To assess the stability of concentrated IPMs as an oral solution in storage condition, the size and LC of IPMs were determined during the storage of the freshly prepared IPMs solution at 4 °C and 25 °C for 8 weeks. As shown in Figure 4, no significant change in the particle size (Figure 4A) and its distribution (Figure 4B), and the LC (Figure 4C) of the IPMs was shown at 4 °C. However, when IPMs were stored at 25 °C, the particle size (Figure 4A) and PDI (Figure 4B) were significantly increased, while the LC was slightly decreased, and no sedimentation was observed in 8 weeks. Therefore, a low temperature storage condition at 4 °C is recommended for the concentrated IPMs.

As reported previously, drug-loaded micelles are generally transported across the intestinal membrane in an intact form [8,31,32]. Therefore, the stability of IPMs in SGF and SIF was important for enhancing the oral bioavailability. The changes in the size and PDI, and LC of concentrated IPMs upon a 25-fold dilution with SGF and SIF at 37 °C are depicted in Figure 4D–F. No remarkable fluctuation in these parameters for IPMs was observed, suggesting that IPMs exhibited favorable dilution-resistant stability in the gastrointestinal tract.

### 3.5. In Vitro Release of IPMs

In vitro release of icaritin from IPMs in SIF and SGF was investigated by dialysis. A dialysis membrane with MWCO 3500 was used to make sure that the release kinetics of icaritin with molecular weight of 368 rather than the membrane-restricted diffusion of icaritin in SGF and SIF was determined under sink conditions [9]. The accumulative release of icaritin as a function of time was plotted in Figure 5. About 19.02 ± 2.03% and 77.28 ± 4.70% of icaritin was released from IPMs within 24 h in SGF and SIF (Figure 5A), respectively, compared to 5.35 ± 2.50% and 16.12 ± 6.71% of icaritin released from the oil suspension (Figure 5B), indicating greatly enhanced and sustained release of icaritin from the polymeric micelles. These might be attributed to the solubility enhancement of icaritin by encapsulation in micelles, and is advantageous for bioavailability improvement after oral administration [29]. Additionally, both the IPMs (Figure 5C) and the oil suspension (Figure 5D) showed a faster release profiles in SIF than in SGF. A possible explanation for such phenomenon might be the better solubility of icaritin in the basic condition.

### 3.6. Bioavailability Studies

As a BCS IV molecule, icaritin was not only poorly water-soluble but also poorly permeable. Therefore, it was a great challenge to improve its oral bioavailability. Nanocrystal formulation has been shown to improve the bioavailability of icaritin by 102% compared with raw icaritin in rats [5]. Moreover, our previous studies demonstrated that the absorption of orally administrated icaritin in humans was similar to that in beagle dogs, and much different from that in mice and rats (unpublished data). Therefore, the pharmacokinetic characters of IPMs were investigated in beagle dogs in this study.

The mean plasma concentration of icaritin versus time profiles are shown in Figure 6, and the pharmacokinetic (PK) parameters are displayed in Table 2. The plasma concentration of icaritin was rapidly decreased after injection (Figure 6A) with a half time (*T*_1/2_) of 5.7 h (Table 2), revealing that the metabolic rate of icaritin was fast and normal formulations for injection were not suitable for the therapy of chronic diseases including cancer.

Furthermore, pharmacokinetic studies were also performed to investigate whether IPMs could enhance the oral bioavailability of icaritin compared with the current clinical formulation (oil suspension) which contained micronized icaritin and corn oil. The plasma concentration of icaritin versus time profiles following oral administration of IPMs and oil suspension to beagle dogs are presented in Figure 6B. The plasma concentration of icaritin for the IPMs treated group was maintained at a high level for a long duration compared with the oil suspension treated group. Notably, two absorption peaks were clearly observed for IPMs, probably due to the reabsorption of icaritin from biliary excretion [33]. The pharmacokinetic parameters are shown in Table 2. The maximum plasma concentration (*C*_max_) and AUC_0–24_ for IPMs were 9.5-fold and 14.9-fold higher, respectively (*p* < 0.01), than those for the oil suspension. In addition, the IPMs reached the *C*_max_ slightly more rapidly than the oil suspension. The absolute bioavailability (*F*%) of the IPMs and the oil suspension were 21.6 ± 13.6% and 1.7 ± 0.5%, respectively.

These findings suggested that the oral absorption of icaritin was considerably enhanced by 14.9-fold with the IPMs in comparison to the oil suspension. This could have resulted from the solubilization, adhesion and inhibition of the P-gp efflux by the polymeric micelles. The enhancement of the solubility of icaritin by the micelles led to the faster drug release (Figure 5) which was essential to developing a high concentration gradient between the drug and the intestinal epithelium, resulting in enhanced permeation effects [29,34]. The favorable adhesion of IPMs of nanoscale size could extend their retention on the intestine epithelial cells, offering more opportunities to transport across the intestinal epithelium [9]. The inhibition of the P-gp efflux by the polymeric micelles could lead to enhancement of the drug’s permeability and absorption to systemic circulation [35]. In addition, the other possible explanation was that the IPMs and oil suspension transported across the intestinal epithelium in different pathways. For the oil suspension, icaritin released from the oil suspension transported across intestinal barrier through passive diffusion. In contrast, the IPMs, being stable in gastrointestinal fluid for at least 12 h (Figure 4D–F), transcytosed across the intestinal barrier through endocytosis as mentioned below [8,31,32]. In summary, the pharmacokinetic study indicated that the Soluplus^®^/P407 micelles could serve as an efficient nanocarrier for icaritin to significantly enhance oral bioavailability.

### 3.7. Transport of IPMs across Caco-2 Cell Monolayers

The above-mentioned results indicated the greatly enhanced bioavailability of icaritin by IPMs with a 14.9-fold increase compared with oil suspension, with higher *C*_max_ and shortened *T*_max_. However, the transport mechanism for the micelles across the intestinal epithelium was not clear. Subsequently, the Caco-2 cell monolayer was used as an epithelial model for evaluating the transcytosis of IPMs, and the micelle integrity following transcytosis and their transcellular transport pathways across Caco-2 cell monolayers were explored in detail.

Firstly, in order to make the FRET micelles retain the physicochemical properties of IPMs as far as possible, a fluorescence probe labelled icaritin (Icaritin-Cy5 and Icaritin-Cy5.5) was synthesized and characterized (Appendix A), and the FRET micelles were prepared by feeding 90% icaritin, 5% Icaritin-Cy5 and 5% Icaritin-Cy5.5. The fluorescence emission spectra of the FRET micelles presented in Figure 7A shows a strong emission at 720 nm by Icaritin-Cy5.5 at an excitation wavelength of 635 nm for Icaritin-Cy5 in water compared with that in methanol, implying that both Icaritin-Cy5 and Icaritin-Cy5.5 were encapsulated in the micelles [8,36]. Hence, FRET micelles were successfully prepared and could be used in the subsequent test.

In order to investigate whether the drug-loaded micelles transported though intestinal membrane as the free drug or in intact micelles, the fluorescence emission spectra of the apical medium and basolateral medium, collected after 4 h of transcytosis of FRET micelles across Caco-2 cell monolayers, were detected at an excitation wavelength of 635 nm. A strong FRET effect was observed in the apical medium (Figure 7B), implying that FRET micelles sustained their intact form, which was supported by the TEM images (Figure 7C). In contrast, a weak FRET phenomenon existed in basolateral medium (Figure 7D), implying that most of the micelles were disaggregated (Figure 7E) and free icaritin-Cy5 and icaritin-Cy5.5 were released. These findings suggested that IPMs were endocytosed by intestinal epithelial cells in their intact forms, and most of them were exocytosed in the form of free icaritin.

In order to clarify the transport pathways involved in the transcellular transport of IPMs, Caco-2 cell monolayers were pre-incubated with endocytosis inhibitors, and the influence of various endocytosis inhibitors on the transport of IPMs was assessed. As was seen from Figure 7F, the amount of transported icaritin was decreased to about 68%, 60%, and 60% of the control in the presence of CMZ, genistein, and amiloride, respectively, implying that macropinocytosis, clathrin-, and caveolae-mediated pathways were involved in the transcellular transport of IPMs. Additionally, the transcytosis of IPMs was enhanced by 135% in the presence of M*β*CD, suggesting that other pathways were probably stimulated when the lipid rafts-mediated pathway was inhibited [37]. Overall, IPMs were transported across intestinal epithelium through multiple pathways.

## 4. Conclusions

In this study, a novel ABS method was developed for the preparation of weakly acidic and alkaline drug-loaded polymeric micelles with high drug loading capacity with water-soluble amphiphilic polymers, and IPMs were successfully prepared via a novel ABS method for oral delivery of icaritin. The prepared IPMs were characterized to be of nanoscale size with a narrow distribution, have favorable storage stability at 4 °C and dilution-resistant stability in GI tract, and enhanced bioavailability compared with oil suspension. Moreover, the transcellular transport pathway of IPMs was demonstrated to include macropinocytosis, clathrin-, and caveolae-mediated pathways. Our findings evidenced the feasibility of the ABS method and Soluplus^®^/P407 micelles for improving oral absorption of icaritin and other poorly water-soluble drugs.

## Data Availability

The data presented in this study are available on request from the corresponding author.

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
