# Peer review of "Enhanced Oral Absorption of Icaritin by Using Mixed Polymeric Micelles Prepared with a Creative Acid-Base Shift Method"

_molecules, 2021, doi:10.3390/molecules26113450_

Round 1
Reviewer 1 Report
The authors described the preparation of mixed polymeric micelles with high drug loading capacity to improve the oral bioavailability of icaritin with Soluplus and Poloxamer 407 using a creative and ingenious acid-base shift method. This article shows interesting results, but more aspects need to be improved before I recommend its publication in Molecules.
- Please give more details in the legend regarding the red, blue and black objects in Scheme 2.
- Please cite in the manuscript the Scheme 1b, 1c and 1d.
- Please mention the manuscript the type of cell for Caco-2 cell line.
- What kind of plate did you use for in vitro experiments? How many wells per plate?
- Why did you use that dose for in vivo administration: Please motivate your choice.
- Do not start the sentences with number values.
- What kind of device did you use to measure TEER values? Please give more details about the device and the measurements.
- How did you prepare the samples from inserts for TEM examination?
- How many times did you repeat the tests? Please include this information in the manuscript.
- Please reformulate the sentence at lines 380, as remdesivir inhibits the virus replication, not virus itself.
- Check the phrase at line 386.
- Check line 402. Is it SEM or TEM?
- How did you conclude that macropinocytosis, clathrin- and caveolae-mediated pathways were involved in transcellular transport of IPMs? Based on what tests?
- The phrase at lines 572-573 regarding the mechanism of transport is not supported by any results from your paper? How do you motivate this phrase?
Author Response
The authors described the preparation of mixed polymeric micelles with high drug loading capacity to improve the oral bioavailability of icaritin with Soluplus and Poloxamer 407 using a creative and ingenious acid-base shift method. This article shows interesting results, but more aspects need to be improved before I recommend its publication in Molecules.
- Please give more details in the legend regarding the red, blue and black objects in Scheme 2.
Thank you for your advices. I have added illustrations of “the red, blue and black objects” to Scheme 2. Red objects and blue objects represent hydrophobic chains and hydrophilic chains of amphiphilic polymers, respectively. Black objects represent drugs.
Scheme 2. An illustration of the different techniques to prepare polymeric micelles. (A) Thin film hydration method. (B) Dialysis method. (C) Simple equilibrium method. (D) Oil in water emulsion method. (E) ABS method. ABS, acid-base shift.
- Please cite in the manuscript the Scheme 1b, 1c and 1d.
Thanks! I have cited the Scheme 1B, 1C and 1D in section “1.introduction”, in the section “2.3. Preparation of Drug-Loaded Polymeric Micelles” and section “3.2. Preparation of Drug-Loaded Micelles by ABS Method”.
- Introduction. In this study, we developed an acid-base shift (ABS) method (Scheme 2E) for preparing icaritin-loaded polymeric micelles (IPMs) based on Soluplus®/Poloxamer 407, and its feasibility was validated with different properties of water-soluble drugs (Scheme 1B, 1C and 1D).
2.3.1. Preparation of Weak Acidic Drugs-Loaded Polymeric Micelles. The weak acidic drugs such as icaritin (Scheme 1A), diosmin (Scheme 1B) and ibuprofen (Scheme 1C) were encapsulated in polymeric micelles by an ABS method.
3.2. Preparation of Drug-Loaded Micelles by ABS Method. The first two drugs with phenolic hydroxyl (Ph–OH, Scheme 1B) or carboxyl group (–COOH, Scheme 1C) are weakly acidic and the last one with amino groups (Scheme 1D) are weakly alkaline.
- Please mention the manuscript the type of cell for Caco-2 cell line.
Thanks for your contribution to our work! I have added the description “The human colon adenocarcinoma cells, Caco-2 cells, at passage 25 were supplied by the Cell Culture Center of Institute of Basic Medical Sciences, Chinese Academy of Medical Sciences and Peking Union Medical College (Beijing, China)” in my manuscript.
- What kind of plate did you use for in vitro experiments? How many wells per plate?
Thanks! Transwell® polycarbonate membrane inserts were used in the in vitro experiments. These inserts are settled in 12-well plates. I have added the description “Caco-2 cell monolayers were obtained by incubation of 500 μL Caco-2 cell suspension (2´105 cells/mL) on Transwell® polycarbonate membrane inserts with pore diameter of 0.4 μm (12-well plates, article number: 3460, Corning, Kennebunk, USA ) in a CO2 incubator for about 21 days” in my manuscript.
- Why did you use that dose for in vivo administration: Please motivate your choice.
Thanks! The effective dose of oil suspension of icaritin in human was 10 mg/kg in current clinical trial[1]. Therefore, 20 mg/kg was used for in vivo experiments according to the conversion factor of equivalent dose, which was about 1.8 times from human to dogs[2] (10 mg/kg × 1.8 = 18 mg/kg). The calculation of absolute bioavailability is closely related when drug exposure is close between oral and i.v. administrations. In explorative tests, it was found that the absolute oral bioavailability of icaritin oil suspension was under 5% in dogs, but IPMs could greatly increase the oral bioavailability of icaritin. So, the i.v. dose of 2 mg/kg was calculated with 10% oral bioavailability (20 mg × 10% = 2 mg/kg).
[1] Fan, Y.; Li, S.; Ding, X.; Yue, J.; Jiang, J.; Zhao, H.; Hao, R.; Qiu, W.; Liu, K.; Li, Y.; et al. First-in-class immune-modulating small molecule Icaritin in advanced hepatocellular carcinoma: preliminary results of safety, durable survival and immune biomarkers. BMC Cancer 2019, 19, 279. DOI:10.1186/s12885-019-5471-1.
[2] FDA Guildance: <<Estimating the Maximum Safe Starting Dose in Initial Clinical Trails for Therapeutics in Adult Healthy Volunteers>>, https://wenku.baidu.com/view/7f23ffa1284ac850ad0242e0.html
- Do not start the sentences with number values.
Your kind suggestion is well received. I have checked all sentences in the manuscript and revised accordingly.
Blood sample (1 mL) for i.v. group was collected from venipuncture of peripheral veins at 0, 0.083, 0.167, 0.5, 1, 2, 4, 6, 8, 12 and 24 h following single administration in heparinized polypropylene tubes.
Blood sample (1 mL) for oral groups was collected from venipuncture of peripheral veins at 0, 0.167, 0.5, 1, 2, 4, 6, 8, 12 and 24 h post-dose in heparinized polypropylene tubes.
- What kind of device did you use to measure TEER values? Please give more details about the device and the measurements.
To answer your question, the following information has been supplemented in the manuscript: “The transepithelial electrical resistance (TEER) was monitored with an electrical resistance meter (Millicell ERS-2, Millipore, Darmstadt, Germany) to detect the integrity of cell monolayers. The monolayers with TEER values of higher than 300 Ω/cm2 were used in the transcytosis tests”.
- How did you prepare the samples from inserts for TEM examination?
The samples were prepared according to the following method: after 4 h of transcytosis, the cell culture medium was collected from the BL side , and a drop of medium without dilution was placed on a carbon-coated copper grid and air dried. The samples were stained with 2% phosphotungstic acid. I have revised the manuscript in “2.4. Characterization of Drug-Loaded Polymeric Micelles”.
- How many times did you repeat the tests? Please include this information in the manuscript.
In this study, particle size and PDI were measured by dynamic light scattering in triplicate; EE%, LC% and content were detected by HPLC in duplicate. DSC, XRD, 1H-NMR and Fluorescence spectrum were tested once. The above information has been supplemented in the manuscript.
- Please reformulate the sentence at lines 380, as remdesivir inhibits the virus replication, not virus itself.
Your suggestion is well received. The revised the sentence “Diosmin is used to treat hemorrhoids[26], ibuprofen is a traditional anti-inflammatory drug, and remdesivir effectively inhibits Ebola virus and 2019-nCov replication”.
- Check the phrase at line 386.
The corrected sentence should write “In addition, to investigate the scale-up effect of ABS method, large scale IPMs (40 L) was produced in the reactor”
- Check line 402. Is it SEM or TEM?
It is SEM at line 402. In the study, IPMs and FRET micelles were observed using SEM and TEM, respectively.
- How did you conclude that macropinocytosis, clathrin- and caveolae-mediated pathways were involved in transcellular transport of IPMs? Based on what tests?
In this study, amiloride, CMZ and genistein were used to inhibit macropinocytosis, clathrin- and caveolae-mediated pathways in Caco-2 cell monolayers[1-4], respectively. The experiments were descripted in Section “2.11. Exploration of Transport Mechanisms of IPMs Across Caco-2 Cell Monolayers”. The influence of various endocytosis inhibitors on the transport of IPMs was assessed. As seen in Figure 7F, the amount of transported icaritin was decreased to about 68%, 60% and 60% of the control in the presence of CMZ, genistein and amiloride, respectively, which indiated that macropinocytosis, clathrin- and caveolae-mediated pathways were involved in transcellular transport of IPMs[1].
Fig.7F The influence of various endocytosis inhibitors on the transcytosis of IPMs across Caco-2 cell monolayers (n = 3). **p < 0.01 and ***p < 0.001 compared to the control.
[1] Qu, X.; Zou, Y.; He, C.; Zhou, Y.; Jin, Y.; Deng, Y.; Wang, Z.; Li, X.; Zhou, Y.; Liu, Y. Improved intestinal absorption of paclitaxel by mixed micelles self-assembled from vitamin E succinate-based amphiphilic polymers and their transcellular transport mechanism and intracellular trafficking routes. Drug Deliv. 2018, 25, 210–225. DOI:10.1080/10717544.2017.1419513.
[2] Ivanov A I, Nusrat A, Parkos C A. Endocytosis of epithelial apical junctional
proteinsby a clathrin-mediated pathway into a unique storage compartment[J]. Mol.
Biol.Cell, 2004(15): 176-188
[3] Hubbard S R, Till J H. Protein tyrosine kinase structure and function[J]. Annu Rev Biochem, 2000(69): 373-398
[4] Orlandi P A, Fishman P H. Filipin-dependent inhibition of cholera toxin: evidence for toxin internalization and activation through caveolae-like domains[J]. J Cell Biol, 1998(141): 905-915
- The phrase at lines 572-573 regarding the mechanism of transport is not supported by any results from your paper? How do you motivate this phrase?
The question maybe answered in the same way as explained in the last question. In this study, amiloride, CMZ and genistein were used to inhibit macropinocytosis, clathrin- and caveolae-mediated pathways in Caco-2 cell monolayers[1-4], respectively. This experiments were descripted in Section “2.11. Exploration of Transport Mechanisms of IPMs Across Caco-2 Cell Monolayers”. The influence of various endocytosis inhibitors on the transport of IPMs was assessed. As seen in Figure 7F, the amount of transported icaritin was decreased to about 68%, 60% and 60% of the control in the presence of CMZ, genistein and amiloride, respectively, which indicated that macropinocytosis, clathrin- and caveolae-mediated pathways were involved in transcellular transport of IPMs[1].
[1] Qu, X.; Zou, Y.; He, C.; Zhou, Y.; Jin, Y.; Deng, Y.; Wang, Z.; Li, X.; Zhou, Y.; Liu, Y. Improved intestinal absorption of paclitaxel by mixed micelles self-assembled from vitamin E succinate-based amphiphilic polymers and their transcellular transport mechanism and intracellular trafficking routes. Drug Deliv. 2018, 25, 210–225. DOI:10.1080/10717544.2017.1419513.
[2] Ivanov A I, Nusrat A, Parkos C A. Endocytosis of epithelial apical junctional
proteinsby a clathrin-mediated pathway into a unique storage compartment[J]. Mol.
Biol.Cell, 2004(15): 176-188
[3] Hubbard S R, Till J H. Protein tyrosine kinase structure and function[J]. Annu Rev Biochem, 2000(69): 373-398
[4] Orlandi P A, Fishman P H. Filipin-dependent inhibition of cholera toxin: evidence for toxin internalization and activation through caveolae-like domains[J]. J Cell Biol, 1998(141): 905-915

Reviewer 2 Report
The manuscript entitled: “Enhanced Oral Absorption of Icaritin by Using Mixed Polymeric Micelles Prepared with a Creative Acid-Base Shift Method” is a very interesting and complete study about the preparation, physicochemical characterization and in vivo biodistribution of Icaritin in mixed polymeric micelles. I recommend the publication of the manuscript after elucidate some details:
- I recommend the authors to confirm the critical micellar concentration by the pyrene fluorescente probe, because is a standardized and more scientifically credible method;
- I recommend the authors to perform zeta potential by Electrophoretic Light Scattering, in order to have more information about the stability of the mixed polymeric micelles;
- I recommend the authors to perform in vitro cytotoxicity studies in fibroblasts due the presence of bases and acids in the final formulations;
Author Response
Comments and Suggestions for Authors
The manuscript entitled: “Enhanced Oral Absorption of Icaritin by Using Mixed Polymeric Micelles Prepared with a Creative Acid-Base Shift Method” is a very interesting and complete study about the preparation, physicochemical characterization and in vivo biodistribution of Icaritin in mixed polymeric micelles. I recommend the publication of the manuscript after elucidate some details:
- I recommend the authors to confirm the critical micellar concentration by the pyrene fluorescente probe, because is a standardized and more scientifically credible method;
In this study, Soluplus®/P407 mixed micelles were developed by referring to Soluplus®/TPGS mixed micelles[1]. In this reference[1], in where CMC was evaluated by ultraviolet spectrophotometry (KI/I2) rather than pyrene fluorescente probe. Therefore, the CMC of Soluplus®/P407 mixed micelles in this study was also evaluated by the ultraviolet spectrophotometry. There are three references[2-4] about CMC detection via pyrene fluorescente probe method, which may guild us to evaluate CMC according to your suggestion.
[1] Ding, Y.; Wang, C.; Wang, Y.; Xu, Y.; Zhao, J.; Gao, M.; Ding, Y.; Peng, J.; Li, L. Development and evaluation of a novel drug delivery: Soluplus®/TPGS mixed micelles loaded with piperine in vitro and in vivo. Drug Dev. Ind. Pharm. 2018, 44, 1409–1416. DOI:10.1080/03639045.2018.1472277
[2] Ray G B , Chakraborty I , Moulik S P . Pyrene absorption can be a convenient method for probing critical micellar concentration (cmc) and indexing micellar polarity[J]. J Colloid Interface, 2006, 294(1):248-254.
[3] Asakawa T , Saruta A , Miyagishi S . Distribution of fluorocarbon quencher among micelles via pyrene fluorescence probe method[J]. Colloid and Polymer Science, 1997, 275(10):958-963.
[4] Zhang, W.; Huang, J.; Fan, N.; Yu, J.; Liu, Y.; Liu, S.; Wang, D.; Li, Y. Nanomicelle with long-term circulation and enhanced stability of camptothecin based on mPEGylated α,β-poly (L-aspartic acid)-camptothecin conjugate. Colloids Surf. B. Biointerfaces 2010, 81, 297–303. DOI:10.1016/j.colsurfb.2010.07.019
- I recommend the authors to perform zeta potential by Electrophoretic Light Scattering, in order to have more information about the stability of the mixed polymeric micelles;
Zeta potential by Electrophoretic Light Scattering has been performed, and the spectrum was shown in Figure 3D, where shows that the zeta potential was nearly neutral (-0.02 mV), because the IPMs consisted with nonionic surfactants (Soluplus® and P407) and a free drug (icaritin). Zeta potential has an important effect on the storage stability of micelles. The absolute Zeta potential value of -30 mV to +30 mV suggests that the particles would remain in a suspended state for a long time[1]. A long term stability of IPMs was further investigated in “2.6. Assessment of the Stability of Micelles” and results showed that IPMs were stable at 4°C but not at 25°C.
Figure 3D Zeta potential of IPMs
[1] Maji R, Dey NS, Satapathy BS, Mukherjee B, Mondal S. Preparation and characterization of Tamoxifen citrate loaded nanoparticles for breast cancer therapy. Int J Nanomedicine. 2014;9(1):3107-3118. https://doi.org/10.2147/IJN.S63535
- I recommend the authors to perform in vitro cytotoxicity studies in fibroblasts due the presence of bases and acids in the final formulations;
Soluplus® and P407 were commercially available excipients and widely used in marketed drugs[1-2]. Therefore, Soluplus® and P407 were safe as oral delivery excipients. The Phase-II and Phase-III clinical trial of icaritin showed there was no ≥grade III drug related AE observed in all HCC patients[3-4]. Although acid and base were used in the process, the final formulation was neutral. Furthermore, impurities of IPMs were investigated and the results showed that each single impurity’s content was less than 0.2%. Thus, the drug degradation induced by acid and base process was considered to be negligible according to International Council for Harmonisation of Technical Requirements for Pharmaceuticals for Human Use (ICH) Q3B. Therefore, the safety risk of IPMs was low. However, we will take your advice to perform in vitro cytotoxicity studies in fibroblasts prior to in vivo safety evaluation.
[1]Soluplus,https://pharmaceutical.basf.com/global/en/drug-formulation/products/soluplus.html.
[2]P407,https://pharmaceutical.basf.com/global/en/drug-formulation/products/kolliphor-p-407.html.
[3] Fan, Y.; Li, S.; Ding, X.; Yue, J.; Jiang, J.; Zhao, H.; Hao, R.; Qiu, W.; Liu, K.; Li, Y.; et al. First-in-class immune-modulating small molecule Icaritin in advanced hepatocellular carcinoma: preliminary results of safety, durable survival and immune biomarkers. BMC Cancer 2019, 19, 279. DOI:10.1186/s12885-019-5471-1.
[4] ASCO 2021 icaritin 4077, https://biotechradar.eu/wp-content/uploads/misc/ASCO21_abstracts_for_Twitter-28-04-2021.pdf

Reviewer 3 Report
The study focuses on the increasing of the oral bioavailability of icaritin by using micelles obtained from Soluplus and Poloxamer 407. Importantly, also the scale-up method was developed. The subject is interesting as well as the obtained results. However, although the micelles have been characterized for their stability, morphology, etc. the biocompatibility issue has been completely ignored. The most important drawback of this paper is lack of in vitro study on cells including biocompatibility and biological activity.
The other minor points:
- Section 2.5 - "... at a wavelength of 273 nm for icaritin, remdesivir, Diosmin and ibuprofen" - the wavelength of each drug should be specified
- Section 2.6 - Isn't the LC a better parameter to study micelles' stability than the EE?
Author Response
The study focuses on the increasing of the oral bioavailability of icaritin by using micelles obtained from Soluplus and Poloxamer 407. Importantly, also the scale-up method was developed. The subject is interesting as well as the obtained results. However, although the micelles have been characterized for their stability, morphology, etc. the biocompatibility issue has been completely ignored. The most important drawback of this paper is lack of in vitro study on cells including biocompatibility and biological activity.
Your advices are sincerely appreciated! In this study, icaritin-loaded polymeric micelles (IPMs) were developed as an oral solution to improve the bioavailability of icaritin. IPMs consisted of Soluplus®, P407 and icaritin. The two major excipients Soluplus® and P407 are commercially available and widely used in marketed drugs[1-2]. Therefore, these excipients are very safe. In addition, the Phase-II and Phase-III clinical trial of icaritin showed there was no ≥grade III drug related AE observed in all HCC patients[3-4]. IPMs as an oral formulation of icaritin was considered to be safe. Thus, biocompatibility of IPMs was ignored. The Phase-II and Phase-III clinical trial of icaritin also showed robust efficacy[3-4]. Therefore, the biological activity of IPMs was also ignored in this study. The drawback of clinical formulation of icaritin (oil suspension) was its low bioavailability (ca. 2%) and our study focused on improving the oral bioavailability of icaritin. Nevertheless, your suggestions are well received, the biocompatibility and biological activity of IPMs are to be evaluated in future studies.
[1]Soluplus,https://pharmaceutical.basf.com/global/en/drug-formulation/products/soluplus.html.
[2]P407,https://pharmaceutical.basf.com/global/en/drug-formulation/products/kolliphor-p-407.html.
[3] Fan, Y.; Li, S.; Ding, X.; Yue, J.; Jiang, J.; Zhao, H.; Hao, R.; Qiu, W.; Liu, K.; Li, Y.; et al. First-in-class immune-modulating small molecule Icaritin in advanced hepatocellular carcinoma: preliminary results of safety, durable survival and immune biomarkers. BMC Cancer 2019, 19, 279. DOI:10.1186/s12885-019-5471-1.
[4] ASCO 2021 icaritin 4077, https://biotechradar.eu/wp-content/uploads/misc/ASCO21_abstracts_for_Twitter-28-04-2021.pdf
The other minor points:
- Section 2.5 - "... at a wavelength of 273 nm for icaritin, remdesivir, Diosmin and ibuprofen" - the wavelength of each drug should be specified
Thanks for your contribution to our work. Content of drug should be determined using HPLC with a UV detector set at a wavelength with high absorption and low disturbance. UV-vis absorption spectrum of icaritin was shown in Figure 1. Icaritin solution at 273 nm exhibited high absorption and low disturbance of solvents and excipients. According to United States Pharmacopeia (USP) and other references, the wavelength of Diosmin, Ibuprofen and Remdesivir were set at 275 nm, 254 nm and 254 nm[1-3], respectively. EE and LC of Diosmin-loaded, Remdesivir-loaded and Ibuprofen-loaded polymeric micelles were determined using a new HPLC method. The results were shown in Table 1 and the manuscript has been revised accordingly.
Figure 1 UV-vis absorption spectrum of icaritin
Table 1. Characteristics of drug-loaded polymeric micelles prepared via ABS method (n = 3).
|
Drug |
Diameter (nm) |
PDI |
EE (%) |
LC (%) |
|
Icaritin (small scale) |
72.74 ± 0.51 |
0.176 ± 0.010 |
92.25 |
13.18 |
|
Diosmin |
98.65 ± 0.75 |
0.096 ± 0.014 |
87.20 |
6.71 |
|
Ibuprofen |
63.95 ± 0.32 |
0.083 ± 0.002 |
90.20 |
6.94 |
|
Remdesivir |
65.20 ± 0.24 |
0.020 ± 0.005 |
97.13 |
14.86 |
|
Icaritin (large scale) |
69.07 ± 0.38 |
0.011 ± 0.018 |
97.40 |
13.92 |
ABS, acid-base shift; EE, encapsulation efficiency; LC, loading content; PDI, polydispersity index.
[1] Diosmin. USP40 –NF35, 6922.
[2] Ibuprofen. USP40-NF35, 4555
[3] Warren, T.K.; Jordan, R.; Lo, M.K.; Ray, A.S.; Mackman, R.L.; Soloveva, V.; Siegel, D.; Perron, M.; Bannister, R.; Hui, H.C.; et al. Therapeutic efficacy of the small molecule GS-5734 against Ebola virus in rhesus monkeys. Nature 2016, 531, 381–385. DOI:10.1038/nature17180.
- Section 2.6 - Isn't the LC a better parameter to study micelles' stability than the EE?
Thanks for your advice! It is true that LC% is a better parameter than EE% to evaluate the stability of micelles. The manuscript has been revised accordingly, and the updated data is shown in Figure 4.
Figure 4. Changes in particle size (A, D), PDI (B, E) and LC (C, F) of IPMs with temperature (A, B, C) and physiological conditions (D, E, F). LC, drug loading content; IPMs, icaritin-loaded polymeric micelles; PDI, polydispersity index.

Round 2
Reviewer 1 Report
The authors addressed all the comments and the manuscript has improved.
Reviewer 3 Report
The manuscript has been corrected according to the Reviewer's comment and may be considered for publication.